# Characterization of the biochemical activity and tumor-promoting role of the dual protein methyltransferase METL-13/METTL13 in *Caenorhabditis elegans*

**Melanie L. Engelfriet, Jędrzej M. Małecki [ID], Anna F. Forsberg [ID], Pål Ø. Falnes\*, Rafal Ciosk [ID]\***

Department of Biosciences, Faculty of Mathematics and Natural Sciences, University of Oslo, Oslo, Norway

\* rafal.ciosk@ibv.uio.no (RC); pal.falnes@ibv.uio.no (PØF)

**Data Availability Statement:** The mass spectrometry proteomics data have been deposited to the ProteomeXchange Consortium via the

## Abstract

The methyltransferase-like protein 13 (METTL13) methylates the eukaryotic elongation factor 1 alpha (eEF1A) on two locations: the N-terminal amino group and lysine 55. The absence of this methylation leads to reduced protein synthesis and cell proliferation in human cancer cells. Previous studies showed that METTL13 is dispensable in non-transformed cells, making it potentially interesting for cancer therapy. However, METTL13 has not been examined yet in whole animals. Here, we used the nematode *Caenorhabditis elegans* as a simple model to assess the functions of METTL13. Using methyltransferase assays and mass spectrometry, we show that the *C. elegans* METTL13 (METL-13) methylates eEF1A (EEF-1A) in the same way as the human protein. Crucially, the cancer-promoting role of METL-13 is also conserved and depends on the methylation of EEF-1A, like in human cells. At the same time, METL-13 appears dispensable for animal growth, development, and stress responses. This makes *C. elegans* a convenient whole-animal model for studying METTL13-dependent carcinogenesis without the complications of interfering with essential wild-type functions.

## Introduction

Protein methylation is a prevalent type of post-translational modification (PTM). It modulates the activity, cellular location, and interactions of proteins. Protein methyltransferases (MTases) are enzymes that catalyze the transfer of a methyl group from the donor *S*-adenosyl-methionine (AdoMet) to a target amino acid residue [1]. The human methyltransferasome contains 208 known or putative MTases, of which most fall in either the seven-β-strand (7BS) family (60% of MTases) or the Su(var)3-9, Enhancer-of-zeste, Trithorax (SET) family (27% of MTases) [2]. Methylation occurs most often on lysines and arginines, but other residues, such as histidines and the N- and C-termini of proteins, are also methylated [3–12]. Lysine residues can receive up to three methyl groups on the ε-nitrogen of the lysine side chain, resulting in their mono-, di-, or trimethylation (referred to here as me1, me2, and me3, respectively).

PRIDE partner repository with the dataset identifier PXD042540 and 10.6019/PXD042540. All other data are within the manuscript and its Supporting Information files.

**Funding:** This work was supported by the Research Council of Norway grant FRIMEDBIO-286499 to RC, and by the Norwegian Cancer Society's grants 228890 to RC and 207855 to PF. The research leading to these results also received funding from the Norwegian Financial Mechanism 2014–2021 operated by the Polish National Science Center under the project contract nr UMO-2019/34/H/NZ3/00691. Some of the strains were provided by the Caenorhabditis Genetics Center (CGC), funded by the NIH. The funders had no role in study design, data collection and analysis, decision to publish, or preparation of the manuscript.

**Competing interests:** The authors have declared that no competing interests exist.

Although lysine methylation of histone proteins has received much attention, many lysine-specific MTases (KMTs) methylate non-histone proteins [4, 9].

The translation of mRNA into protein, which depends on a number of conserved translation factors, consists of three steps: initiation, elongation, and termination [13]. The eukaryotic elongation factor 1 alpha (eEF1A) functions during translation elongation to deliver an aminoacyl-tRNA to the A-site of the translating ribosome as an aminoacyl-tRNA-eEF1A-GTP complex. Complementary tRNA-mRNA codon recognition stimulates eEF1A to hydrolyze GTP and causes eEF1A-GDP to exit the complex.

Mammalian eEF1A contains multiple reported PTMs, of which phosphorylation and methylation have been accredited to influencing its activity in translation to various degrees. Phosphorylation of multiple residues on eEF1A by different kinases stimulates the overall protein translation, with the exception of phosphorylated S300, which blocks aminoacyl-tRNA binding [14–17]. The methylation landscape of human eEF1A includes five methylated lysine residues and the methylated N-terminus [18]. Although little is known about the methylation status of the eEF1A N-terminus in relation to its function, methylation of the different lysine residues seems to modulate translation rates of different codons or proteins involved in various cellular processes [18]. The KMTs responsible for methylating human eEF1A at the various lysine residues were identified, with eEF1A-KMT4 (ECE2), eEF1A-KMT1 (N6AMT2), eEF1A-KMT3 (METTL21B), and eEF1A-KMT2 (METTL10) being responsible for methylating K36, K79, K165, and K318, respectively [19–23]. The newest player here is the methyltransferase-like protein 13 (METTL13). Its sole substrate appears to be the translation factor eEF1A [24, 25]. METTL13 is a dual 7BS MTase, containing an N-terminal KMT domain which mediates dimethylation of K55 and a C-terminal MTase domain which, after removal of the initiator methionine by methionine aminopeptidases, trimethylates the N-terminal amino group of G2 [24, 25]. The methylation marks deposited by METTL13 seem to fine-tune the activity of eEF1A in translation [24].

METTL13 and dimethylated K55 on eEF1A are abundant in many human cancers, which correlates with poor patient prognosis [25–28]. Recent studies have strengthened the cancer connection, as proliferation and global protein synthesis are impaired in transformed cells depleted of METTL13 [25, 26]. Furthermore, the loss of METTL13 function in xenograft mouse models, and depletion of pancreatic METTL13 in a mouse model of Ras-driven pancreatic duct adenocarcinoma (PDAC), were shown to suppress tumor growth [25, 26]. However, it is not clear what roles, if any, METTL13 plays in wild-type animals. Here, we examined the possible functions of METTL13 in a whole animal, the nematode *C. elegans*. Combining biochemical and genetic approaches, we show that the *C. elegans* METTL13, METL-13, methylates the nematode eEF1A, EEF-1A, at the same positions as the human protein. Examining animals lacking METL-13 or its methylations of EEF-1A, we observed no evident defects in animal growth and development. However, we found that the METL-13-dependent methylation of EEF-1A promotes *C. elegans* tumorigenesis. We conclude that METTL13 has a conserved biochemical function from humans to nematodes. Remarkably, METL-13/METTL13 and its methylation of EEF-1A/eEF1A appear to be dispensable for animals under laboratory culture conditions. Thus, assuming the conservation, targeting METTL13 in cancer therapy could bypass the toxicity associated with targeting essential proteins.

## Results

### *C. elegans* METL-13 is a dual-specificity methyltransferase targeting the G2 and K55 residues of EEF-1A

The *C. elegans* ortholog of the human *METTL13* gene is C01B10.8, henceforth *metl-13*. The human METTL13 and nematode METL-13 proteins are highly similar: 33% of their amino

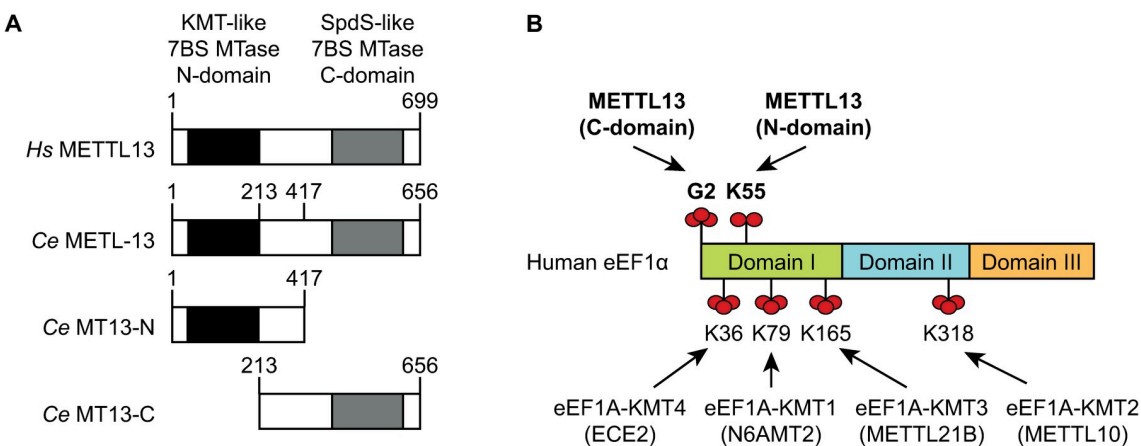

**Fig 1. The domain organization of METTL13 is conserved in *C. elegans*. A.** Overview of the domain organization of human METTL13 and *C. elegans* METL-13. The boundaries of MTase domains (MT13-C and MT13-N), used in the *in vitro* methyltransferase assays, are indicated by amino acid positions. **B.** Architecture of the human eEF1A protein and its methylations. Arrows point to residues methylated by the indicated MTases, with METTL13 and its targeted residues in bold. Red spheres indicate methyl groups.

acid sequence is identical and they share similar domain arrangement, with two 7BS MTase-like domains separated by a linker region (**Fig 1A**). The N-terminal KMT domain of METTL13 dimethylates eEF1A at K55 (eEF1A$^{K55me2}$), whereas the C-terminal KMT domain trimethylates the amino group of G2 (eEF1A$^{G2me3}$) [24] (**Fig 1A and 1B**).

To find out if nematode METL-13 has similar enzymatic activities as its human counterpart, we expressed and purified from *E. coli* two N-terminally hexahistidine (His$_6$) tagged fragments of METL-13 encompassing its individual MTase domains (**Fig 1A**). These METL-13 fragments were tested in an *in vitro* assay to monitor their putative MTase activities, using recombinant human (*Hs*) eEF1A as the protein substrate and [$^3$H]-AdoMet, as methyl donor, allowing the detection of [$^3$H]-methylated eEF1A by fluorography as a distinct band on SDS-PAGE. We decided to use the recombinant *Hs* eEF1A as the substrate based on its high homology to the nematode ortholog and nearly identical sequence surrounding the putative METL-13 target sites (**S1A Fig in S1 File**). We observed that the C-terminal domain of METL-13 could methylate eEF1A bearing a C-terminal His-tag, but not when the His-tag was present on the N-terminus, suggesting that the methylation occurs at the N-terminus of eEF1A (**Fig 2A**). Indeed, mutating G2 to I (G2I) abolished the methylation of eEF1A, whereas mutating K55 to R (K55R) had little effect (**Fig 2A**). In contrast, the N-terminal domain of METL-13 methylated *Hs* eEF1A, regardless of the position of the His-tag or the G2I substitution (**Fig 2B**). Most importantly, the methylation was abolished by the K55R substitution, pointing to K55 of eEF1A as the key methylation site (**Fig 2B**). These experiments show that the enzymatic activities of individual domains of METL-13 match the established activities of human METTL13, i.e., its N-terminal domain targets K55, whereas the C-terminal domain targets G2 of eEF1A.

To demonstrate that METL-13 functions as a MTase *in vivo*, we used a previously uncharacterized mutant strain, carrying a homozygous deletion in the *metl-13* locus. This mutation, *metl-13(tm6870)*, causes a frameshift at the end of exon 3, leading to a premature stop codon (**S1B Fig in S1 File**). Nonsense-mediated mRNA decay (NMD) is expected to degrade the resulting mRNA and, indeed, the levels of *metl-13* mRNA were significantly lower in the *metl-13(tm6870)* strain than in wild-type (wt) (**Fig 2C**). Moreover, any residual METL-13 protein would lack part of the N-terminal MTase domain and the whole C-terminal MTase domain.

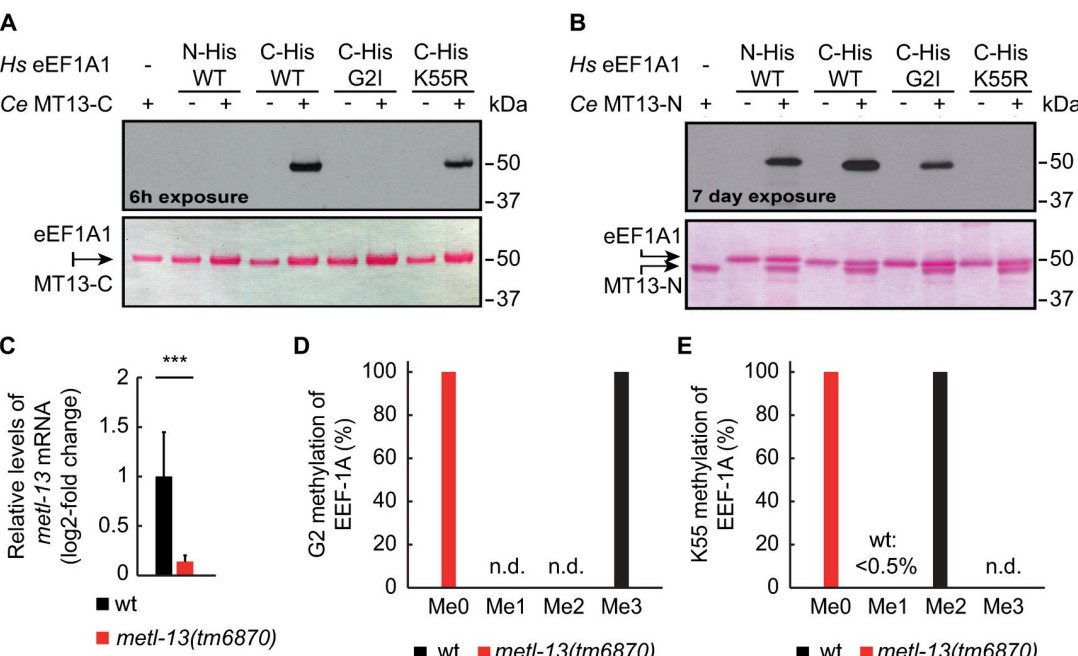

**Fig 2. The biochemical function of METTL13 is conserved in *C. elegans*. A,B.** Evaluation of the methyltransferase activity of the *C. elegans* METL-13 fragments, using recombinant human eEF1A1 as the substrate. The MT13-C (A) and MT13-N (B) fragments were incubated with [³H]-AdoMet and human eEF1A1 variants as indicated; His-tagged at either the N- or C-terminus and in some cases with point substitutions (G2I or K55R). Methylation was visualized by fluorography (top panels). Protein loading was assessed by Ponceau S staining (bottom panels). **C.** RT-qPCR analysis of *metl-13* mRNA levels in wild-type (wt) and *metl-13(tm6870)* animals. The levels of *metl-13* were assessed using primers spanning the exon 2–3 junction and normalized to the levels of *act-1* (actin) mRNA. Error bar indicates SEM from 3 biological replicates; ***: p<0.001. **D,E.** Quantitative analysis of the endogenous methylation of EEF-1A, on G2 (D) and K55 (E) in the extracts from wt or *metl-13 (tm6870)* animals, using LC-MS/MS; n.d.: not detected.

Thus, *metl-13(tm6870)* animals should express no functional METL-13 protein. To confirm that the *metl-13(tm6870)* strain is devoid of METL-13 activity, we analyzed the methylation status of G2 and K55 of the endogenous EEF-1A/eEF1A in *metl-13(tm6890)* and wild-type animals. Using liquid chromatography-tandem mass spectrometry (LC-MS/MS), we found that G2 on EEF-1A was fully trimethylated and K55 was virtually fully (99.6%) dimethylated in wild-type animals (**Fig 2D and 2E**). In contrast, both residues were completely unmethylated in *metl-13(tm6870)* mutants (**Fig 2D and 2E**). Thus, METL-13 is responsible for the G2 and K55 methylation of EEF-1A *in vivo*.

Next, since the main target of METTL13 is eEF1A, we examined the potential impact of METL-13-mediated methylation on EEF-1A. Firstly, we examined the levels of EEF-1A by western blotting but observed no difference in its levels between *metl-13(tm6870)* mutants and wild-type animals (**Fig 3A**). Secondly, we used CRISPR-Cas9 engineering to create the K55R substitution in EEF-1A. Since the highly similar genes *eef-1A.1* and *eef-1A.2* (94% identical covering 1392 bases, including a stop codon) encode the fully identical EEF-1A protein, we first created the *eef-1A.1(syb2837)* and *eef-1A.2(syb2785)* single mutants, each encoding EEF-1A containing a K55R mutation. These two mutants were then crossed to obtain the *eef-1A.1 (syb2837); eef-1A.2(syb2785)* double mutant. Analyzing this strain, we observed no difference between the levels of wild-type and K55R EEF-1A (**Fig 3A**).

Finally, using surface sensing of translation (SUnSET) [29, 30], we asked whether the loss of METL-13 impacts global protein synthesis. However, we observed no obvious difference between the wild type and *metl-13(tm6870)* mutants (**Fig 3B**). Concluding, the biochemical

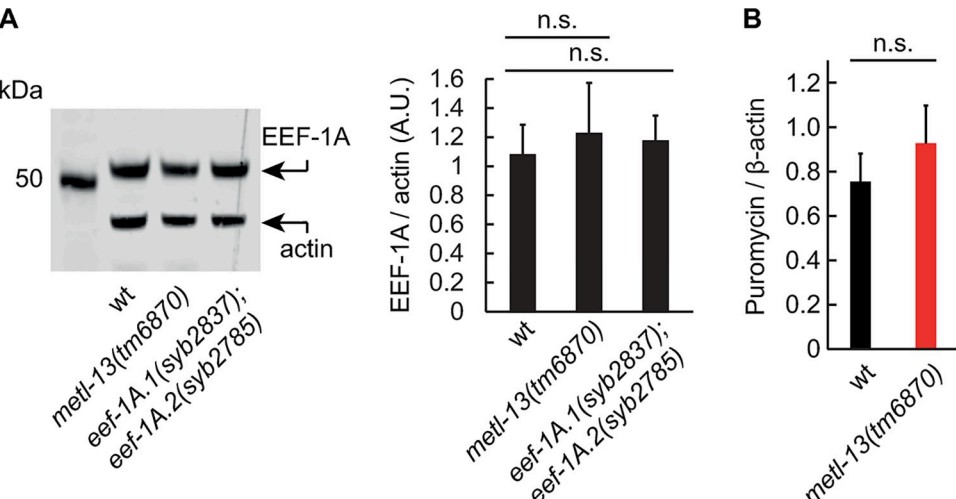

**Fig 3. Ablation of METL-13 or EEF-1A$^{K55me2}$ does not affect protein levels of EEF-1A or protein synthesis. A.** Protein levels of the *C. elegans* eEF1A, EEF-1A, do not change upon the loss of METL-13. Left: western blot showing the levels of EEF-1A and actin (loading control) in whole lysates from mixed-staged wt, *metl-13(tm6870)*, and *eef-1A.1* (syb2837); *eef-1A.2*(syb2785) mutant animals. Right: Densiometric quantification of EEF-1A levels, normalized to actin. Error bars indicate SEM of 3 biological replicates; n.s.: not significant. **B.** Quantification of puromycin incorporation in the lysates from wt and *metl-13(tm6870)* mutants. The levels of incorporated puromycin were normalized to the levels of β-actin used as a loading control. Error bars indicate SEM of 3 biological replicates; n.s.: not significant.

activity of *C. elegans* METL-13 and human METTL13 is conserved. METTL13/METL-13 targets eEF1A/EEF-1A at two independent sites, introducing dimethylation at K55 and trimethylation at G2. However, at standard growth conditions, these modifications appear to have no obvious impact on global translation.

## The K55 methylation of EEF-1A increases the penetrance of germline tumors

In humans, increased levels of METTL13 and eEF1A$^{K55me2}$ correlate with poor survival of patients with lung or pancreatic cancer [25]. Furthermore, the ablation of METTL13 reduces proliferation and global protein synthesis in cancer cells, while non-transformed cells appear unaffected [25, 26]. Since the biochemical function of METL-13 is conserved in *C. elegans*, we tested whether it also plays a role in tumorigenesis. Many *C. elegans* factors function as germline tumor suppressors [31]. Among them is GLD-1, a maxi-KH/STAR RNA-binding protein functioning as a translational repressor [32–35]. Another example is PRO-1, a conserved WD-repeat protein implicated in rRNA processing [36]. Without these factors, germ cells develop into highly penetrant tumors, although found in different gonadal regions (**Fig 4A**).

Therefore, to evaluate the role of METL-13 in tumorigenesis, we knocked down *gld-1* by RNAi in either wild-type animals or *metl-13(tm6870)* mutants. Upon depletion of GLD-1, germ cells that have entered meiosis revert to the mitotic cycle, resulting in an ectopic mass of proliferative cells that we refer to as a 'fully proliferative' phenotype. In the wild type, most gonads displayed large tumors in the proximal/medial regions, as expected; these tumors consisted of small, undifferentiated cells (**Fig 4B**). In the *metl-13(tm6870)* mutants, however, many of the tumorous gonads contained enlarged cells reminiscent of differentiating oocytes, we refer to this gonad phenotype as 'proliferative with enlarged cells' (**Fig 4B and S2A Fig in S1 File**). Importantly, we observed a similar increase in the differentiating cells in the *eef-1A.1*

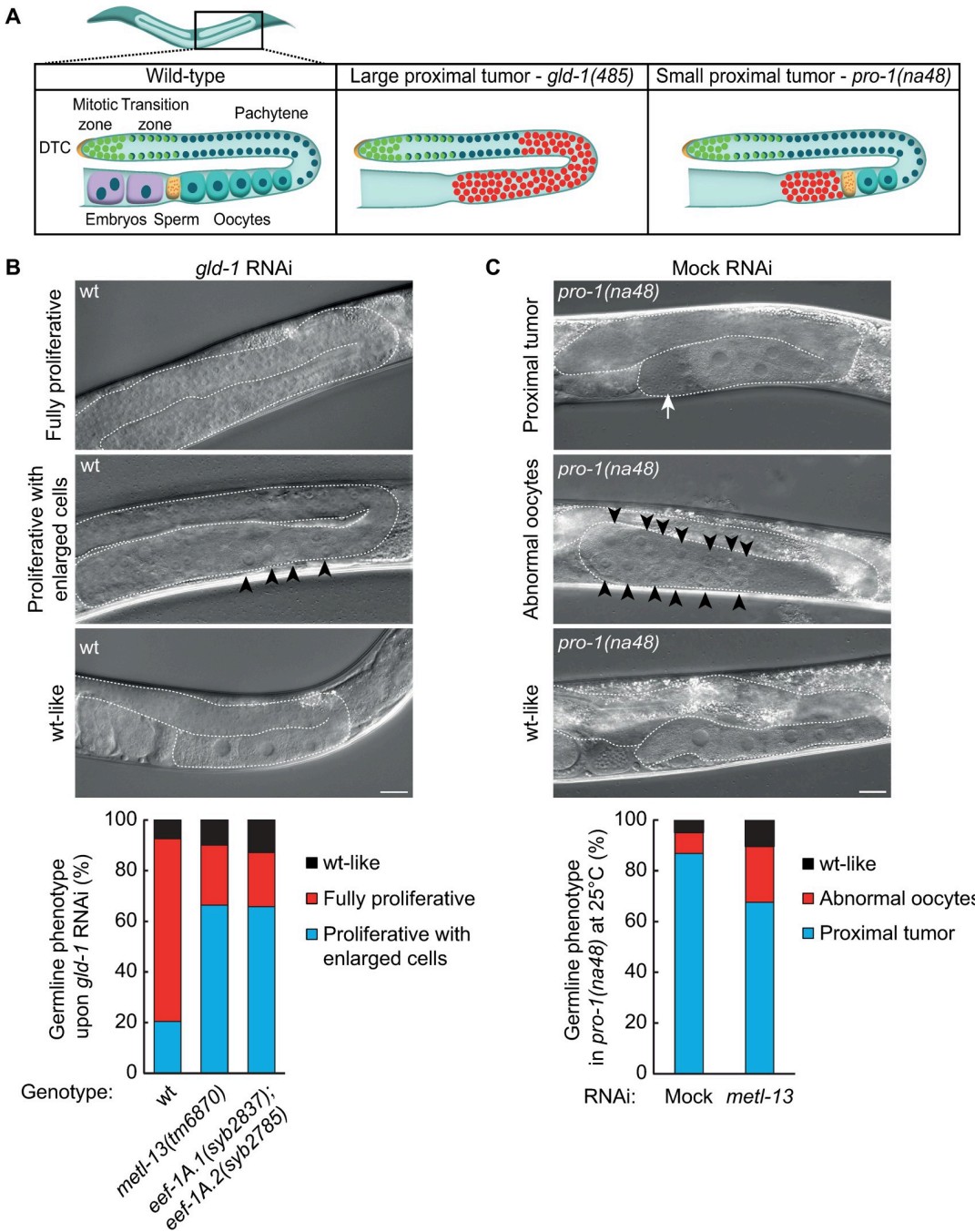

**Fig 4. Depletion of METL-13 reduces the severity of germline tumors. A.** Diagrams of wild-type and tumorous *C. elegans* germlines, tumorous cells are marked in red. While the gonads of gld-1(q485) mutants display large proximal germline tumors, the gonads of *pro-1(na48)* mutants grown at the restrictive temperature (25˚C) contain smaller tumors located proximal to the gametes. **B.** DIC micrographs of adult wt animals subjected to *gld-1* RNAi. The gonads are outlined. Three phenotypes were observed after *gld-1* RNAi, we refer to these gonads as 'fully proliferative', 'proliferative with enlarged cells' (black arrowheads indicate enlarged cells), and 'wt-like' respectively. Scale bar = 20 μm. Quantification of the germline phenotypes observed after *gld-1* RNAi in wt, *metl-13(tm6870)*, and *eef-1A.1(syb2837); eef-1A.2(syb2785)* animals show that the predominant phenotype is 'fully proliferative' in wt and 'proliferative with enlarged cells' in *metl-13(tm6870)* and *eef-1A.1(syb2837; eef-1A.2(syb2785)* animals. The quantification shows the average phenotype distribution from three biological replicates, 70 germlines were assessed per biological replicate. **C.** DIC micrographs of *pro-1(na48)* mutants (grown at the restrictive temperature, 25˚C). The gonads are outlined. Three phenotypes were observed: we refer to seemingly normal germlines as 'wt-like', germlines containing proliferating cells proximal to the gametes as 'proximal tumor' (indicated by a white arrow), and germlines lacking a proximal tumor that instead contain abnormal oocytes as 'abnormal oocytes' (indicated by black arrowheads). Scale

bar = 20 µm. Quantification of the germline phenotypes observed after mock or *metl-13* RNAi show a decrease in the 'proximal tumor' phenotype and an increase in the 'abnormal oocytes' and 'wt-like' phenotypes when *metl-13* is knocked-down. The quantification shows the average phenotype distribution from three biological replicates, 70 germlines were assessed per biological replicate.

*(syb2837); eef-1A.2(syb2785)* double mutants (**Fig 4B** and **S2A Fig in S1 File**). These differences were not caused by less efficient RNAi in the absence of METL-13-mediated methylation, as RNAi against another target was as efficient in the *metl-13(tm6870)* or *eef-1A.1 (syb2837); eef-1A.2(syb2785)* mutants as in the wild-type (**S3A Fig in S1 File**). These results suggest that the tumors are less penetrant without the K55 methylation of EEF-1A. To confirm this finding in another tumorous background, we RNAi-depleted *metl-13* in temperature-sensitive *pro-1(na48)* mutants. At the restrictive temperature (25˚C), most of the *pro-1(na48)* gonads displayed proximal tumors, while a minority looked superficially wild-type or contained abnormal oocytes (**Fig 4C**). However, the numbers of gonads with proximal tumors significantly decreased upon *metl-13* RNAi, and the fraction of wt-like gonads/gonads containing abnormal oocytes increased (**Fig 4C** and **S2B Fig in S1 File**). Concluding, our results suggest that METL-13-mediated methylation of EEF-1A enhances tumorigenesis in the nematodes, which is reminiscent of the tumor-promoting role of human METTL13.

## METL-13 is non-essential for animal growth and development

To analyze the potential function of METL-13 in wild-type animals, we singled synchronized L4 wild-type or *metl-13(tm6870)* animals to individual plates, allowed them to grow, and counted their offspring. Since the speed of *C. elegans* development and the offspring size depend on temperature [37], these experiments were performed at 15˚C, 20˚C, and 25˚C (**Fig 5A**). At all tested temperatures, we observed no difference in the offspring or developmental speed between the wild-type and mutant animals (**Fig 5B**).

Since tumorigenesis is linked to increased protein synthesis, we also examined animals cultivated on bacterial diet enhancing growth and development. In contrast to a standard diet of the *E. coli* OP50 strain, animals fed the *E. coli* HB101 strain develop faster and grow larger [38–41]. This is due to an increased cell size rather than cell number and reflects the increased protein content of HB101-fed animals [40]. To further accelerate growth and thus the demand for protein synthesis, the animals were cultured at 25˚C [37]. As anticipated, wild-type animals that were fed HB101 bacteria developed faster and grew bigger. However, there was no difference between the wild-type and *metl-13(tm6870)* mutants (**Fig 5C and 5D**). Thus METL-13 appears dispensable for wild-type growth and development, even under conditions requiring increased protein synthesis.

Finally, when comparing the lifespan of animals with or without functional METL-13, we noticed a small but significant difference, with the *metl-13(tm6870)* mutants dying a little faster than the wild-type (**Fig 6**). However, the lifespan of *eef-1A.1(syb2837); eef-1A.2(syb2785)* double mutants was not different from the wild-type (**Fig 6**), suggesting that the longevity-related role of METL-13 is unrelated to the K55 methylation of EEF-1A.

Lifespan assay performed at 20˚C, in 3 biological replicates (>100 animals per strain, per replicate). P-value wt vs. *metl-13(tm6870)*: 0.024; p-value wt vs. *eef-1A.1(K55R); eef-1A.2 (K55R)*: 0.43, log-rank test.

## Discussion

Here, we have studied the previously uncharacterized *C. elegans* orthologue METL-13. The biochemical function of METL-13 is remarkably conserved, with the dual methyltransferase

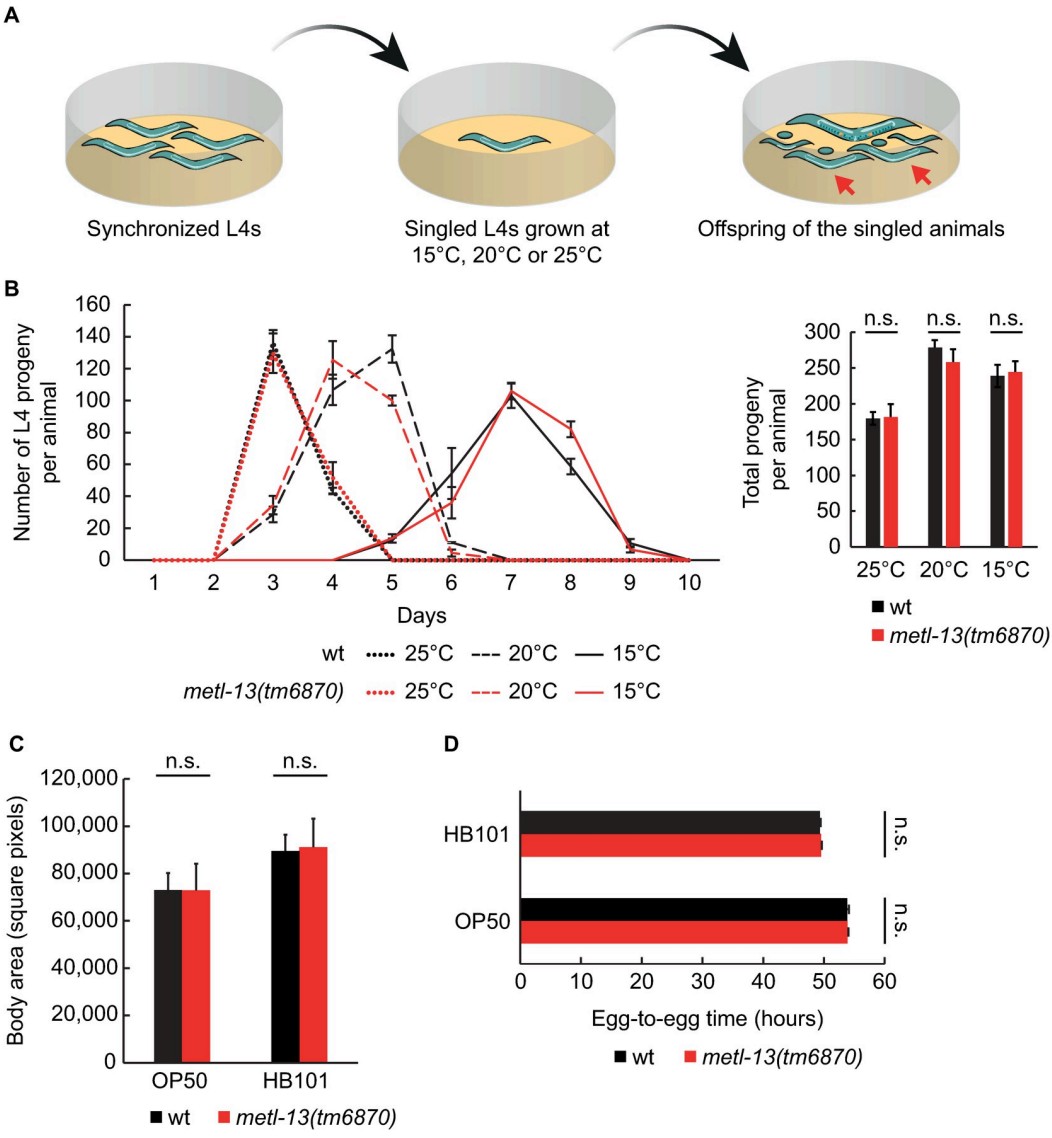

**Fig 5. The loss of METL-13 has no obvious impact on the *C. elegans* growth under standard culture conditions or nutrient-rich diet. A.** Schematic representation of the development and fecundity assays shown in B. Synchronized wt and *metl-13(tm6870)* animals were grown to the L4 stage at 20°C, after which they were singled to individual plates and grown at indicated temperatures. Their offspring were counted when reaching the L4 stage (red arrows) and removed from the plate. **B.** Left: The L4 offspring of singled animals were counted each day at the indicated temperatures. Note that they developed into L4s with a similar speed as wild type. Error bars indicate SEM from 3 biological replicates. Right: Total L4 offspring per animal at the indicated temperatures. Error bars indicate SEM from 3 biological replicates. **C.** Body area measurements of 1-day old adult wt and *metl-13(tm6870)* animals grown at 25°C on a diet of OP50 or HB101 bacteria. Error bars indicate SEM of 3 biological replicates, the body area of a minimum of 15 worms was measured per biological replicate. **D.** Developmental time of wt and *metl-13(tm6870)* nematodes grown from an egg until an egg-laying adult at 25°C, on a diet of OP50 or HB101 bacteria. Error bars indicate SEM of 3 biological replicates, developmental egg-to-egg time was measured for 20 worms per biological replicate.

being responsible for the same methylation marks *in vivo* on the nematode EEF-1A/eEF1A as in humans.

There is much interest in METTL13 due to its apparent role in cancer. METTL13 and eEF1A$^{K55me2}$ are upregulated in a plethora of human cancers [25–28]. Moreover, targeting METTL13 or the downstream eEF1A$^{K55}$ inhibits neoplastic growth in human cell lines, as well

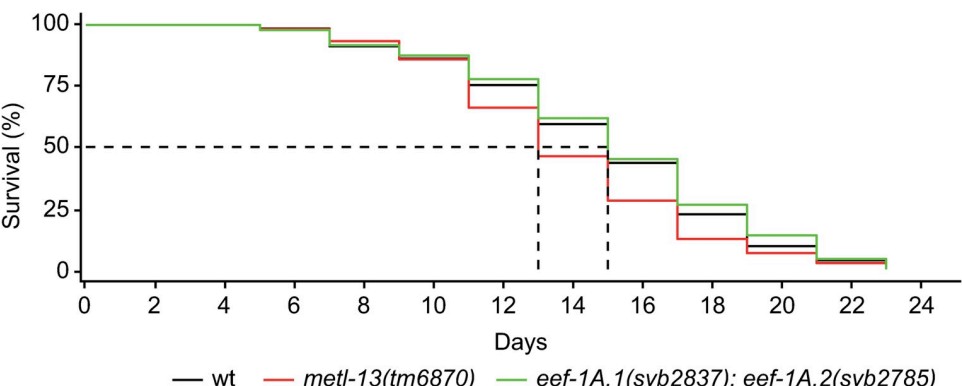

**Fig 6. The absence of METL-13 impacts lifespan independently of the EEF-1A K55 methylation.**

as in xenograft and PDAC mouse models [25, 26]. Remarkably, the depletion of METL-13 has a similar inhibitory effect on the *C. elegans* tumors, as the severity of genetically induced germline tumors declined in the absence of METL-13 or EEF-1A$^{K55me2}$. We note that inhibiting METL-13 or EEF-1A$^{K55me2}$ did not suppress tumors completely. Also, the partial suppression was no longer observed when we used the *gld-1(q485)* null mutants instead of *gld-1* RNAi (**S3B Fig in S1 File**). Thus, the loss of EEF-1A$^{K55me2}$ reduces but does not abolish cancer, at least in the tested tumorous genetic backgrounds.

So how does eEF1A$^{K55me2}$ impact cancer? The methylation of K55 upregulates the GTPase activity of eEF1A, which may cause increased protein synthesis needed for tumorigenic growth [25]. The elongation factor eEF1A is known to be present in excess compared to other components of the translation machinery. This suggests that eEF1A is unlikely to be rate-limiting in protein synthesis and that its non-canonical functions might instead play a role in cancer [42, 43]. Alternatively, translation capacity may be in surplus under normal conditions but becomes limiting in cancer. This is consistent with the finding that eIF4E haploinsufficient mice develop normally, with wild-type levels of protein synthesis, but are more resistant to oncogenic transformation [44]. While the etiology of human and *C. elegans* tumors is likely different, it is intriguing that mutating various *C. elegans* RNA-binding proteins, which like GLD-1 function to repress translation, leads to germline tumors [31]. Thus, reducing translational output, by removing METL-13, could explain the partial suppression of the tumorous proliferation observed upon the depletion of GLD-1. Whether the same applies to *pro-1* tumors is less clear, as their etiology is less well understood [36].

Cancer treatments typically involve the inhibition of essential proteins like cell cycle regulators and signaling molecules. This is why chemotherapy often causes serious adverse effects in cancer patients [45–47]. The METTL13-mediated methylation of eEF1A could be inhibited by small molecules. However, METTL13 knock-outs have thus far only been studied in cell lines and specific tissues [25, 26]. Our whole-animal study suggests that a complete removal of METL-13 or EEF1-A$^{K55me2}$ has no obvious impact on wild-type growth and development. This does not preclude potential functions under specific conditions or contexts. Curiously, METL-13 appears to play a limited but significant role in longevity. However, animals in which EEF-1A is no longer methylated on K55 do not display this longevity defect. Thus, this effect could depend on METTL13-mediated methylation of the G2 residue on EEF-1A, the methylation of another substrate, or an additional function of METTL13. In summary, our studies establish a simple model for METL-13-assisted tumorigenesis and endorse the eEF1A$^{K55}$ modification as an attractive target in cancer therapy, whose inhibition could help

**Table 1. Strains used in this study.**

| Strain genotype | Source | Identifier |
|---|---|---|
| Wild type | CGC | N2 |
| *metl-13(tm6870) (IV)* | NBRP | RAF2267 |
| *eef-1A.1(syb2837) (III); eef-1A.2(syb2785) (X)* | This study | RAF2260 |
| *pro-1(na48)/mIn1[dpy-10(e128) mIs14] (II)* | CGC | GC565 |
| *metl-13(tm6870) (IV); lin-41(rrr3) (I); xeSi197[Plin-41::flag::gfp::lin-41::lin-41 3'UTR; unc-119(+)] (II)* | This study | RAF2265 |
| *eef-1A.1(syb2837) (III); eef-1A.1(syb2785) (X); lin-41(rrr3) (I); xeSi197[Plin-41::flag::gfp::lin-41::lin-41 3'UTR; unc-119(+)] (II)* | This study | RAF2266 |

reduce cancer growth without the side effects associated with traditional drugs used in chemotherapy.

## Methods

### *C. elegans* culture and generation of mutant strains

Animals were maintained using standard procedures and grown at 20˚C unless stated otherwise [41]. Information on the strains used in this study are reported in **Table 1**. The *metl-13 (tm6870)* strain was outcrossed thrice with wild-type males to reduce background mutations. The *eef-1A.1(syb2837)* and *eef-1A.2(syb2785)* strains were generated by SunyBiotech using CRISPR-Cas9 technology and outcrossed thrice with wild-type males before crossing them together to generate the double mutant, which was again outcrossed three-times with wild-type males to reduce background mutations. To synchronize animals, embryos were extracted from gravid adults by treatment with a bleaching solution (30% (v/v) sodium hypochlorite (5% chlorine) reagent (ThermoFisher Scientific; 419550010), 750 mM KOH). The embryos were left to hatch overnight into arrested L1 larvae in the absence of food, at RT in M9 buffer (42 mM $Na_2HPO_4$, 22 mM $KH_2PO_4$, 86 mM NaCl, 1 mM $MgSO_4$). For RNAi experiments, synchronized L1s were grown at 25˚C to adults on RNAi-inducing NGM agar plates seeded with HT115 *E. coli* bacteria containing plasmids targeting genes of interest [48].

### Cloning of METL-13

The ORFs encoding fragments of METL-13 protein (MT13-N: amino acids 1–417; MT13-C: amino acids 213–656), were amplified by PCR from *C. elegans* cDNA and cloned between NdeI and NotI sites into pET28a plasmid (Novagen), using In-Fusion® HD Cloning Plus kit (Takara). The cloning primers are listed in **Table 2**. Sanger sequencing was used to verify all cloned constructs.

### Expression and purification of recombinant proteins

Expression and purification of recombinant N-terminally $His_6$-tagged *C. elegans* MT13-N and MT13-C proteins from *E. coli* was performed using Ni-NTA-agarose (Qiagen), following a

**Table 2. Cloning primers used in this study.**

| Name | Sequence (5'→ 3') |
|---|---|
| MT13N-NdeI-F | CGCGGCAGCCATATGTCCGCTCCAAATGAGC |
| MT13N-NotI-R | CTCGAGTGCGGCCGCTTATTTCAGATAAGCTTCAGATTGTAC |
| MT13C-NdeI-F | CGCGGCAGCCATATGCCTCTAGAAGTTCTTCGCTCC |
| MT13C-NotI-R | CTCGAGTGCGGCCGCTTAATAATCAACCATACGAATGTTTGAG |

previously described protocol [49]. Cloning, mutagenesis and purification of recombinant human eEF1A1, containing either N-terminal or C-terminal His$_6$-tag, was described previously [21, 24]. Protein concentration was determined using Pierce BCA Protein Assay Kit (Thermo Fisher Scientific). Recombinant proteins were stored in Storage Buffer (50 mM Tris-HCl pH 7.4, 100 mM NaCl, 10% glycerol) in single use aliquots, kept at −20˚C.

### *In vitro* methyltransferase assay using [$^3$H]AdoMet

MTase activity assay using *C. elegans* MT13-N or MT13-C as enzymes, human eEF1A1 as the substrate and [$^3$H]AdoMet (PerkinElmer) as the methyl donor, was performed similarly as previously described [24]. 10 µl reactions were set-up on ice in Storage Buffer, containing 1 µg of eEF1A1, 1 µg of MT13-N or MT13-C and 0.5 µCi of [$^3$H]AdoMet. Reaction mixtures were incubated for 1 h at 30˚C and analyzed by SDS-PAGE. Proteins were transferred to polyvinylidene difluoride membrane and stained with Ponceau S. Incorporation of tritium-labeled methyl groups into proteins was visualized by fluorography. All methyltransferase experiments were independently replicated at least two times.

### RNA extraction

Pellets of 100 µl containing a mix of embryos, larvae and adults were prepared by harvesting the animals (grown on NGM agar plates with OP50 bacteria), adding 250 µl Trizol and snap-freezing in liquid nitrogen. Animals were lysed by 10 freeze-thaw cycles with liquid nitrogen and incubating at 37˚C respectively. Chloroform extraction was used by adding 50 µl chloroform and centrifugation at 12000 x g for 10 minutes. The aqueous phase was transferred into new microcentrifuge tubes and 1 volume of isopropanol was added and mixed well. The samples were centrifugated at 16100 x g for 20 minutes at 4˚C, after which the RNA pellet was washed with 80% ethanol and air-dried before being resuspended in nuclease-free water. The quality of the RNA was assessed using NanoDrop Spectrophotometer.

### RT-qPCR

The QuantiTect Reverse Transcription Kit (Qiagen) was used for elimination of genomic DNA and reverse transcription using 1 microgram of RNA. The cDNA was diluted 1:5 and 2.5 µl of this dilution was used with 2 µl HOT FIREPol EvaGreen qPCR Mix (Solis BioDyne, 08-36-00001) and 0.25 µl of 10 µM gene-specific forward and reverse primers for *metl-13* (forward: CGGAATTAGCGACCCAACTCT; reverse: TCGGCCATGGATAGATTAGCA) or *act-1* (forward: TTGCCCCATCAACCATGAAGA; reverse: TGTGCAAGTTGACGAAGTTGTG).

### Enrichment of EEF-1A from nematode extracts

Pellets containing a mix of embryos, larvae and adults were prepared by collecting the animals (grown on NGM agar plates with *E. coli* OP50 bacteria) in M9 buffer and washing thrice before removing all of the buffer and snap-freezing in liquid nitrogen. Extracts were prepared by grinding the pellet with a mortar and pestle in the presence of liquid nitrogen and dissolving in Lysis Buffer (50 mM Tris-HCl pH 7.4, 100 mM NaCl, 1% Triton X-100, 5% glycerol (w/vol), 5 mg/ml cOmplete Proteinase Inhibitor Tablets (Roche, 11697498001), 1% P8340 (Sigma)). Extracts were cleared by centrifugation at 16100 x g for 5 minutes at 4˚C. EEF-1A protein was enriched from the *C. elegans* extracts using a method previously employed for enrichment of eEF1A1 present in human cell extracts [21]. Extracts were loaded on Pierce Strong Cation Exchange (S) Spin Columns (Thermo Fisher Scientific, 90008) pre-equilibrated in Lysis Buffer, followed by extensive washing of the columns with Washing Buffer (50 mM Tris-HCl pH 7.4,

150 mM NaCl, 5% glycerol (w/vol), 5 mg/ml cOmplete Proteinase Inhibitor Tablets (Roche, 11697498001), 1% P8340 (Sigma)). The retained material, including EEF-1A, was eluted from the columns with Elution Buffer (50 mM Tris-HCl pH 7.4, 400 mM NaCl, 5% glycerol (w/vol), 5 mg/ml cOmplete Proteinase Inhibitor Tablets (Roche, 11697498001), 1% P8340 (Sigma)).

## Mass spectrometry

Mass spectrometry analysis was essentially performed as previously described [24]. In brief, protein samples enriched for EEF-1A (obtained as described above) were resolved by SDS-PAGE and stained with Coomassie Blue. A band corresponding to the mass of EEF-1A (~ 50 kDa) was excised from the gel and subjected to in-gel digestion using chymotrypsin or Glu-C, after which the resulting proteolytic fragments were desalted using 10 μl OMIX C18 micro-SPE pipette tips (Agilent, Santa Clara, CA, USA). The samples were analyzed by nanoLC-MS using either an Ultimate 3000 RSLCnano-UHPLC system connected to a Q Exactive mass spectrometer (Thermo Fisher Scientific, Bremen, Germany) or a NanoElute-UHPLC coupled to a TimsTOF Pro mass spectrometer (Bruker Daltonics, Bremen, Germany). The LC-MS data were analyzed using Peaks Studio version 10.6 (Bioinformatics Solution Inc.), searching against the *C. elegans* protein sequence database from Swiss-Prot (4155 entries). The maximum number of variable modifications detected on a peptide was restricted to three, allowing the following variable modifications: methionine oxidation, cysteine propionamidation, methylation of lysine (mono, di, and tri), and methylation of the N-terminus (mono, di, and tri). MS/MS spectra of peptides containing the N-terminal G2 or K55 of EEF-1A were inspected for presence of methylated residues, followed by quantitation of G2 and K55 methylation status using PEAKS Studio. The mass spectrometry proteomics data have been deposited to the ProteomeXchange Consortium via the PRIDE partner repository with the dataset identifier PXD042540 and 10.6019/PXD042540 [50].

## Western blot analysis

Pellets containing a mix of embryos, larvae and adults were prepared as described above. Lysates were prepared by grinding the pellet with a mortar and pestle in the presence of liquid nitrogen and dissolving in lysis buffer (50 mM HEPES (pH 7.4), 150 mM KCl, 5 mM MgCl$_2$, 0.1% Triton X-100, 5% glycerol (w/vol), 1 mM PMSF, 7 mg/ml cOmplete Proteinase Inhibitor Tablets (Roche, 11697498001). Lysates were cleared by centrifugation at 16100 x g for 20 minutes at 4˚C. Protein concentrations were approximated by Bradford Assay (Bio-Rad). Samples were prepared by mixing 25 micrograms of protein extract with the required amount of 4x NuPAGE™ LDS Sample Buffer (Invitrogen, NP0007) and 10x NuPAGE Sample Reducing Agent (Invitrogen, NP0004), followed by heating at 70˚C for 10 minutes. Proteins were separated by SDS-PAGE and transferred onto a polyvinylidene difluoride membrane by wet-transfer. The following primary antibodies were used: 1:1000 monoclonal mouse anti-eEF1A (Merck, 05–235), 1:10000 monoclonal mouse anti-puromycin (Merck, MABE343), 1:5000 polyclonal rabbit anti-actin (Abcam, ab8227). Detection was carried out with IRDye 680RD-conjugated goat anti-mouse secondary antibody (LI-COR Biosciences, 926–68070) or IRDye 800CW-conjugated goat anti-rabbit secondary antibody (LI-COR Biosciences, 926–32211) and infrared imaging (LI-COR Biosciences, Odyssey CLx).

## SUnSET assay

Synchronous arrested L1 larvae were obtained as described above. A total of 16000 L1 larvae were grown until the young adult stage on NGM plates seeded with *E. coli* OP50 bacteria. Animals were washed, grown and treated with puromycin as previously described [29]. For the

Western blot analysis, three micrograms of total protein were used per well. Incorporation of puromycin into nascent peptides was measured by normalizing band intensity from monoclonal anti-puromycin (Merck, MABE343) antibody to anti-actin (Abcam, ab8227).

## Microscopy

Animals were transferred into a drop of 10 mM levamisole on a 2% (w/v) agarose pad, covered with a cover slip and immediately imaged with a Zeiss AxioImager Z1 microscope. Images were acquired with an Axiocam MRm REV2 CCD camera using the Zen software (Zeiss). The acquired images were processed with Image J.

## Development and fecundity assay

Synchronized arrested L1 larvae were plated on NGM agar plates with OP50 and grown at 25˚C for 24 hours to the L4 larval stage. Individual animals were picked over so that there is one single worm per NGM agar plate. The animals and their offspring were left to grow at 25˚C, 20˚C, or 15˚C. The parental animal was picked over to a new plate every 24 hours and the offspring was counted and removed from the plate when they reached the L4 stage.

## Body area measurements

Synchronous arrested L1 larvae were obtained as described above. A minimum of 50 animals were grown at 25˚C until 1-day old adults on NGM plates seeded with either *E. coli* OP50 or HB101 bacteria. Micrographs depicting the whole animal were taken of a minimum of 15 worms for each strain and each food-source condition per biological replicate. Body area of the animals in square pixels was measured with Image J.

## Egg-to-egg assay

Four adult animals were picked on an NGM plate seeded with either *E. coli* OP50 or *E. coli* HB101 bacteria to lay eggs at 25˚C for 2 hours before being removed. The eggs were grown at 25˚C into the L4 stage before separating a single worm per plate for a total of 20 worms for each strain and food-source condition per biological replicate. The animals were kept at 25˚C and checked every hour after entering adulthood to mark the time at which they start laying their first egg.

## Lifespan assay

Synchronous arrested L1 larvae were obtained as described above. A minimum of 100 L1 larvae were plated out per NGM agar plate and left to grow until the young adult stage at 20˚C. Dead animals were counted and removed from the plate every 48 hours. The population of animals was moved to a new plate every 24 hours during fertile adulthood to omit the offspring from the assay.

## Supporting information

**S1 File. Contains all the supporting figures.**
(DOCX)

**S1 Raw images.**
(PDF)

## Author Contributions

**Conceptualization:** Pål Ø. Falnes, Rafal Ciosk.

**Formal analysis:** Melanie L. Engelfriet, Jędrzej M. Małecki, Anna F. Forsberg, Pål Ø. Falnes, Rafal Ciosk.

**Funding acquisition:** Pål Ø. Falnes, Rafal Ciosk.

**Investigation:** Melanie L. Engelfriet, Jędrzej M. Małecki, Anna F. Forsberg, Rafal Ciosk.

**Methodology:** Melanie L. Engelfriet, Jędrzej M. Małecki, Rafal Ciosk.

**Project administration:** Rafal Ciosk.

**Resources:** Rafal Ciosk.

**Supervision:** Pål Ø. Falnes, Rafal Ciosk.

**Validation:** Melanie L. Engelfriet, Jędrzej M. Małecki, Rafal Ciosk.

**Visualization:** Melanie L. Engelfriet, Jędrzej M. Małecki, Rafal Ciosk.

**Writing – original draft:** Melanie L. Engelfriet, Jędrzej M. Małecki, Pål Ø. Falnes, Rafal Ciosk.

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
