## [Decision Letter · Decision Letter 0]

28 Apr 2023

PONE-D-23-07410Characterization of the biochemical activity and tumor-promoting role of the dual protein methyltransferase METL-13/METTL13 in Caenorhabditis elegans.PLOS ONE

Dear Dr. Ciosk,

Thank you for submitting your manuscript to PLOS ONE. After careful consideration, we feel that it has merit but does not fully meet PLOS ONE’s publication criteria as it currently stands. Therefore, we invite you to submit a revised version of the manuscript that addresses the points raised during the review process. The only additional experiment that needs to be performed, is to repeat the experiment in Figure 4B with the expression of wildtype proteins to rule out that the phenotypes observed are due to undetected background mutations. Other changes that are requested by the reviewers can be addressed by modifying the figures and text.

We look forward to receiving your revised manuscript.

Kind regards,

Alexander F. Palazzo, Ph.D.

Academic Editor

PLOS ONE

Reviewers' comments:

Reviewer's Responses to Questions

**Comments to the Author**

1. Is the manuscript technically sound, and do the data support the conclusions?

Reviewer #1: Yes

Reviewer #2: Partly

2. Has the statistical analysis been performed appropriately and rigorously? 

Reviewer #1: Yes

Reviewer #2: Yes

3. Have the authors made all data underlying the findings in their manuscript fully available?

Reviewer #1: Yes

Reviewer #2: No

4. Is the manuscript presented in an intelligible fashion and written in standard English?

Reviewer #1: Yes

Reviewer #2: Yes

5. Review Comments to the Author

Reviewer #1: The authors describe the in vivo relevance of an uncharacterized methyltransferase family member METTL13 using C. elegans as the model. METTL13 in humans is shown to methylate two sites on the translation elongation factor, but its in vivo relevance in an organismic model is not known. Here they demonstrate the biochemical activity of worm METTL13, show that it is dispensable in vivo, but show that germline tumors depend on it for full penetrance. This is a highly relevant finding, given that human protein and methylation of the translation elongation factor is linked to tumors. This raises the possibility that genetic replacement of the worm protein with the human orthologue may allow creation of a genetic model system for screening small molecule inhibitors. I find this study to be very interesting and support its publication.

Minor:

Some of the references linked to protein methyltransferases could include the more recent ones.

Please mention what the other modifications on the elongation factor does for its in vivo functions.

A clearer description of the tumor phenotype (fully proliferative vs fully proliferative with enlarged cells) will be useful.

Reviewer #2: In the manuscript by Engelfriet et al, the authors use C. elegans as a model to study the methyltransferase METL-13 and its methylation of EEF-1A (homologues of the human METTL13 and eEF1A respectively). METTL13 and the methylation of eEF1A have been previously shown to promote carcinogenesis in cell culture, but an in vivo/whole animal study has not been conducted yet.

First the authors show that the enzymatic activity of the two METL-13 MTase domains is the same between humans and worms by using the worm MTAse domains to methylate recombinant human eEF1A. eEF1A is methylated at the same positions by the worm MTases showing strong functional homology even though the sequences diverged over the course of evolution, which strengthens the author’s point of C. elegans being able to serve as an appropriate model for the role in human cancer. The authors show that the in vivo methylation of EEF-1A at positions G2 and K55 is solely dependent on the presence of METL-13 by looking at the methylation pattern in a metl-13 mutant. The authors further investigate the role of METL-13 and methylation of EEF-1A in two germline tumor contexts. Both the metl-13 mutant and a double-mutant eef-1A strain (which lacks the K55 methylation sites) show a decreased severity of tumors in a gld-1 knockdown, and a knockdown of metl-13 shows a lowered tumor burden in a temperature sensitive pro-1 mutant. The authors note that absence of METL-13 or the methylation sites in EEF-1A does not fully prevent tumorigenesis, but still reduces tumor burden in both contexts. In order to show that an absence of METL-13 is not detrimental to the fitness of worms (and thereby a potentially good target for therapeutics), the authors performed experiments showing that fertility and development of worms is not affected in a metl-13 mutant strain.

With this study the authors show that C. elegans can serve as a model to further study METTL13’s role in carcinogenesis and development of potential treatment options based on targeting of METTL13. The manuscript is overall well written, easy to follow, and the conclusions drawn by the authors are supported by the presented experiments and results. Although I already consider this an excellent manuscript, I have a few points that should be addressed before being accepted for publication to bolster the author’s conclusions, which I will detail below.

Major points:

Experiment presented in Fig. 4B: In order to fully support the conclusion that a mutation in either metl-13 or the K55 methylation sites of eef-1A are responsible for the decreased severity of the tumors, it would be necessary to reconstitute the mutants by expressing the wildtype proteins. By showing that addition of the wildtype proteins restores the wildtype phenotype, the authors can strengthen their conclusions (and exclude undetected background mutations).

Fig 4B/C: The authors should also include the sample size of observations on which the bar graphs are based on. Furthermore, the authors should strongly consider showing example images from gld-1 RNAi treated mutant worms and pro-1(na48) worms after metl-13 RNAi to allow the reader to assess the tumors in these contexts.

Fig. 4A: Recolor the tumorous cells to have a different color from the early germline cells and reference this color code in the legend or figure itself.

General:

The manuscript is lacking details/methods on the generation of the eef-1A mutant strains.

Can the authors add a section in the discussion to detail in what way the Ce germline tumors are known to be similar or different to the human tumors in which METTL-13 has been shown to be relevant? This would allow the reader to more thoroughly assess how transferable knowledge from the worm model is to human cancer biology.

The authors should deposit the data of their MS experiments in a public repository to allow for analysis by the community.

Minor points:

Fig. 5: Add the number of worms that have been used to the figure legend (in addition to the mention in the methods)

Fig. S1A: I would suggest adding a consensus sequence highlighting the common and different amino acids below the shown human and Ce sequences.

6. PLOS authors have the option to publish the peer review history of their article (what does this mean?). If published, this will include your full peer review and any attached files.

Reviewer #1: No

Reviewer #2: No

---

## [Author Response · Author response to Decision Letter 0]

2 Jun 2023

We appreciate the reviewers’ comments on the manuscript. Below is our point-by-point response to their concerns. 

Reviewer #1: 

Minor:

Some of the references linked to protein methyltransferases could include the more recent ones.

We have included four more recent references to give a better overview of the extent of protein methylation on different types of residues (Cornett et al., Molecular Cell, 2019; Wu et al., Nature Reviews Drug Discovery, 2021; Diaz et al., Current Opinion in Chemical Biology, 2021; Davydova et al., Nature Communications, 2021). The references are linked to the following sentence in the introduction: “Methylation occurs most often on lysines and arginines, but other residues, such as histidines and the N- and C-termini of proteins, are also methylated.”. The more recent review on protein lysine methylation outside of histones (Cornett et al., Molecular Cell, 2019) has also been added as a reference to the following sentence in the introduction: “Although lysine methylation of histone proteins has received much attention, many lysine-specific MTases (KMTs) methylate non-histone proteins.”.

Please mention what the other modifications on the elongation factor does for its in vivo functions.

The introduction section now includes a general description of the effects of phosphorylation and methylation on the canonical function of eEF1A in translation elongation. We hope the Reviewer will agree that further elaborations on different types of PTMs and their effects on canonical and non-canonical functions of eEF1A are outside the scope of this paper. 

A clearer description of the tumor phenotype (fully proliferative vs fully proliferative with enlarged cells) will be useful.

We have added the following statement in the Results section, discussing the tumorous phenotype upon knockdown of gld-1: “Upon depletion of GLD-1, germ cells that have entered meiosis revert to the mitotic cycle, resulting in an ectopic mass of proliferative cells that we refer to as a ‘fully proliferative’ phenotype.”. We have also modified the text in the same paragraph, describing the ‘proliferative with enlarged cells’ phenotype seen in the metl-13(tm6870) and eef-1A.1(syb2837); eef-1A.2(syb2785) mutant animals.

Reviewer #2: 

Major points:

Experiment presented in Fig. 4B: In order to fully support the conclusion that a mutation in either metl-13 or the K55 methylation sites of eef-1A are responsible for the decreased severity of the tumors, it would be necessary to reconstitute the mutants by expressing the wildtype proteins. By showing that addition of the wildtype proteins restores the wildtype phenotype, the authors can strengthen their conclusions (and exclude undetected background mutations).

We understand that such rescuing experiments are common in many fields. However, they are non-trivial and time-consuming to perform in C. elegans. We would either need to revert the mutant strains back to wild-type by CRISPR-Cas9 genome editing or express full-length rescue constructs, including uncharacterized endogenous promoters and 3’UTRs. In the latter case, we would need to insert ectopic copies into a different locus, which also could impact expression. We understand the concern that off-target mutations in the CRISPR-Cas9 edited mutant strains could play a role in the severity of the observed tumor phenotypes. However, we think that undetected background mutations are unlikely to play a role. In this model, it is customary to outcross CRISPR-Cas9 edited strains several times (usually three times) against the wild type to exclude off-target mutations. Indeed, it was done for all mutant strains generated in this study. This is now described in the Methods section (C. elegans culture and generation of mutant strains). We acknowledge that an off-target mutation could occur close to the edited locus and remain in the strain despite outcrossing. However, we find it highly unlikely that two random off-target mutations (one linked to the metl-13 locus and the other to either the eef-1A.1 or eef-1A.2 locus) would cause identical phenotypes, i.e. the suppression of the gld-1 tumor. Thus, we find it justified to conclude that the phenotype is caused by mutations in met-13 or eef-1A.1; eef-1A.2, rather than background mutations. Finally, our observations on the role of METL-13 in C. elegans are in line with the findings reported in mice on METTL13. Considering all that, we hope that the Reviewer will agree that additional control experiments are unnecessary.

Fig 4B/C: The authors should also include the sample size of observations on which the bar graphs are based on. Furthermore, the authors should strongly consider showing example images from gld-1 RNAi treated mutant worms and pro-1(na48) worms after metl-13 RNAi to allow the reader to assess the tumors in these contexts.

The number of germlines that were assessed for each biological replicate is now added to the legend of Figures 4B and 4C. Example images of metl-13(tm6870) and eef-1A.1(syb2837); eef-1A.2(syb2785) mutant animals after gld-1 RNAi are now shown in Supplementary Figure 2A, example images of pro-1(na48) mutant animals after metl-13 RNAi are shown in Supplementary Figure 2B. 

Fig. 4A: Recolor the tumorous cells to have a different color from the early germline cells and reference this color code in the legend or figure itself.

The tumorous cells in the schematic of Figure 4A have been recolored red, which is now also mentioned in the legend of Figure 4A. 

General:

The manuscript is lacking details/methods on the generation of the eef-1A mutant strains.

The single-mutant strains eef-1A.1(syb2837) and eef-1A.2(syb2785) were generated by SunyBiotech using CRISPR/Cas9 genome editing, after which we outcrossed the single-mutants thrice to wild-type N2 animals. After this, the single-mutants were crossed to generate the double-mutant strain eef-1A.1(syb2837); eef-1A.2(syb2785) which was again outcrossed thrice. This is now better explained in the Methods section describing the generation of mutant strains. 

Can the authors add a section in the discussion to detail in what way the Ce germline tumors are known to be similar or different to the human tumors in which METTL-13 has been shown to be relevant? This would allow the reader to more thoroughly assess how transferable knowledge from the worm model is to human cancer biology.

We have now included the following text in the Discussion section: While the etiology of human and C. elegans tumors is likely different, it is intriguing that mutating various C. elegans RNA-binding proteins, which like GLD-1 function to repress translation, leads to germline tumors (Vanden Broek et al., Frontiers in Cell and Developmental Biology, 2022). Thus, reducing translational output, by removing METL-13, could explain the partial suppression of the tumorous proliferation observed upon the depletion of GLD-1. Whether the same applies to pro-1 tumors is less clear, as their etiology is less well understood (Voutev et al., Developmental Biology, 2006).

The authors should deposit the data of their MS experiments in a public repository to allow for analysis by the community.

The MS proteomics data on the methylation status of G2 and K55 in wild-type and metl-13(tm6870) animals has now been deposited to the ProteomeXchange Consortium via the PRIDE partner repository. The dataset identifiers are now described in the text in the Methods section. 

Minor points:

Fig. 5: Add the number of worms that have been used to the figure legend (in addition to the mention in the methods).

We have now added the number of worms used for the body area measurements and the egg-to-egg assays in the legend of Figures 5C and 5D.

Fig. S1A: I would suggest adding a consensus sequence highlighting the common and different amino acids below the shown human and Ce sequences.

Rather than introducing a consensus sequence, we have modified Supplementary Figure S1A by introducing an orange text color to highlight differences in amino acid sequence between the human and C. elegans eEF1A/EEF-1A.

---

## [Editor Report · Decision Letter 1]

7 Jun 2023

Characterization of the biochemical activity and tumor-promoting role of the dual protein methyltransferase METL-13/METTL13 in Caenorhabditis elegans.

PONE-D-23-07410R1

Dear Dr. Ciosk,

We’re pleased to inform you that your manuscript has been judged scientifically suitable for publication and will be formally accepted for publication once it meets all outstanding technical requirements.

Kind regards,

Alexander F. Palazzo, Ph.D.

Academic Editor

PLOS ONE
---

## [Editor Report · Acceptance letter]

13 Jun 2023

PONE-D-23-07410R1 

Characterization of the biochemical activity and tumor-promoting role of the dual protein methyltransferase METL-13/METTL13 in *Caenorhabditis elegans*. 

Dear Dr. Ciosk:

I'm pleased to inform you that your manuscript has been deemed suitable for publication in PLOS ONE. Congratulations! Your manuscript is now with our production department. 

Kind regards, 

on behalf of

Dr. Alexander F. Palazzo 

Academic Editor

PLOS ONE